# Computational Risk Stratification of Preclinical Alzheimer’s in Younger Adults

**DOI:** 10.3390/diagnostics15111327

**Published:** 2025-05-26

**Authors:** Oriehi Anyaiwe, Nandini Nataraj, Bhargava Sai Gudikandula

**Affiliations:** Department of Mathematics and Computer Science, College of Arts and Sciences, Lawrence Technological University, Southfield, MI 48075, USA; nnataraj@ltu.edu (N.N.); bgudikand@ltu.edu (B.S.G.)

**Keywords:** Alzheimer’s disease, cognitive impairment, biomarkers, machine learning, computational modeling, age-specific risk, precision medicine, sigmoid simulation

## Abstract

**Background:** Alzheimer’s disease (AD) is a progressive neurodegenerative disorder that often begins decades before clinical symptoms manifest. Early detection remains critical for effective intervention, particularly in younger adults, where biomarker deviations may signal pre-symptomatic risk. This research presents a computational modeling framework to predict cognitive impairment progression and stratify individuals into risk zones based on age-specific biomarker thresholds. **Methods:** The model integrates sigmoid-based data generation to simulate non-linear biomarker trajectories reflective of real-world disease progression. Core biomarkers—including cerebrospinal fluid (CSF) amyloid-beta 42 (Aβ42), amyloid positron emission tomography (amyloid PET), cerebrospinal fluid Tau protein (CSF Tau), and magnetic resonance imaging with fluorodeoxyglucose positron emission tomography (MRI FDG-PET)—were analyzed simultaneously to compute the cognitive impairment (CI) score of instances, dynamically adjusted for age. Higher CSF Aβ42 levels consistently demonstrated a protective effect, while elevated amyloid PET and Tau levels increased cognitive risk. Age-specific CI thresholds prevented the overestimation of risk in younger individuals and the underestimation in older cohorts. To demonstrate its applicability, we applied the full four-stage framework—comprising data aggregation and cleaning, sigmoid-based synthetic biomarker simulation with descriptive analysis, parameter accumulation modeling, and correlation-driven CI classification—on a curated dataset of 307 instances (ages 10–110) from Kaggle, the Alzheimer’s Disease Neuroimaging Initiative (ANDI), and the Open Access Series of Imaging Studies (OASIS) to evaluate age-specific stratification of preclinical AD risk. **Results:** The study highlights the model’s potential to identify individuals in risk zones from a pool of 150 instances, enabling targeted early interventions. Furthermore, the framework supports retrospective disease trajectory analysis, offering clinicians insights into optimal intervention windows even after symptom onset. **Conclusions:** Future work aims to validate the model using longitudinal, inclusive, real-world datasets and expand its predictive capacity through machine learning techniques and integrating genetic and lifestyle factors. Ultimately, this research contributes to advancing precision medicine approaches in Alzheimer’s disease by providing a scalable computational tool for early risk assessment and intervention planning.

## 1. Introduction

Alzheimer’s disease (AD) is a progressive neurodegenerative disorder characterized by impaired memory, cognitive decline, and diminished capacity to perform daily activities, representing the leading cause of dementia among the elderly. According to the Alzheimer’s Association’s 2024 Facts and Figures report [1], approximately 7 million Americans currently suffer from AD, impacting one in nine individuals aged 65 and older. By 2050, the number of affected individuals is projected to rise to 12.7 million, barring any significant medical breakthroughs in disease prevention or cure.

The diagnosis of AD has traditionally been based on clinical evaluation, comprising a detailed history, cognitive testing, and exclusion of other causes of dementia, formalized in the National Institute of Neurological and Communicative Disorders and Stroke–Alzheimer’s Disease and Related Disorders Association (NINCDS-ADRDA) criteria [2] and later updated by the National Institute on Aging–Alzheimer’s Association (NIA-AA) research framework [3]. These guidelines stratify individuals into “probable” or “possible” AD based on insidious onset and progressive impairment in memory and other cognitive domains, complemented by neuropsychological assessments such as the Mini-Mental State Examination (MMSE) and the Clinical Dementia Rating (CDR) scale [4]. However, clinical diagnosis alone often lacks specificity, particularly in early or preclinical stages, where symptoms overlap with normal aging and other neurodegenerative conditions can lead to misclassification [5,6].

Over the past decade, the integration of fluid and imaging biomarkers has revolutionized AD diagnosis, enabling the detection of underlying pathology long before overt dementia. Core cerebrospinal fluid (CSF) biomarkers—reduced amyloid-beta 42 (Aβ42) and elevated total Tau (t-Tau) or phosphorylated Tau (p-Tau)—provide high diagnostic accuracy for amyloid plaque and neurofibrillary tangle burden [7,8]. Positron emission tomography (PET) ligands targeting amyloid (e.g., Pittsburgh Compound B) and Tau (e.g., Flortaucipir) allow in vivo visualization of hallmark lesions, while fluorodeoxyglucose positron emission tomography [18F]FDG-PET reveals characteristic patterns of cortical hypometabolism in temporoparietal regions [9,10]. Structural MRI further contributes by quantifying hippocampal atrophy and cortical thinning—changes that correlate with progression from mild cognitive impairment (MCI) to dementia [11].

The 2018 NIA-AA “ATN” framework formally classifies individuals by amyloid (A), Tau (T), and neurodegeneration (N) biomarker status, fostering a shift from syndromic to biologically defined AD [12]. This biomarker-driven paradigm promises earlier and more precise identification of at-risk individuals, guiding clinical trials and paving the way for targeted interventions in the preclinical phase of AD.

Presently, there is no cure for AD, and current therapeutic strategies offer limited efficacy in halting the disease progression, providing only temporary symptomatic relief. Extensive research has identified multiple primary risk factors associated with AD, including advanced age, genetics, lifestyle factors, head trauma, cardiovascular conditions, cognitive engagement, and chronic neuronal inflammation [13,14,15]. Notably, emerging evidence indicates that brain atrophy and pathologic changes begin several decades before clinical symptoms become apparent [16,17], underscoring the critical need for early interventions to delay or potentially mitigate the disease’s trajectory [18,19,20].

Due to the complexity of AD pathology, no single biomarker currently provides a definitive indication of cognitive risk or disease progression. Reliance on a solitary biomarker often leads to misclassification, as each captures only specific aspects of the disease mechanism. For example, low cerebrospinal fluid (CSF) Aβ42 levels can indicate amyloid plaque formation, yet similar decreases occur in cognitively normal aging [8,21,22]. Likewise, amyloid PET imaging quantifies plaque burden but does not consistently correlate with present cognitive impairment [10,23]. CSF Tau and phosphorylated Tau (p-Tau) levels reflect neurodegeneration but are less sensitive indicators in early disease stages [10,24]. FDG-PET imaging, which captures brain glucose hypometabolism, typically detects downstream metabolic changes following amyloid and Tau pathology; consequently, it is traditionally considered less sensitive for early-stage detection [10,25].

However, recent advancements highlight that magnetic resonance imaging (MRI) and fluorodeoxyglucose positron emission tomography (FDG-PET) imaging have significant potential in the early diagnosis of AD. MRI offers non-invasive visualization of structural brain changes, such as hippocampal atrophy, cortical thinning, and ventricular enlargement, potentially detectable even before overt clinical symptoms appear [10,26]. FDG-PET, by detecting altered glucose metabolism indicative of synaptic dysfunction and neuronal injury, provides valuable insights into early metabolic changes that precede cognitive decline, enhancing diagnostic accuracy and prognosis [27,28]. Combining MRI with FDG-PET could further improve early detection by simultaneously capturing complementary anatomical and metabolic biomarkers, thereby offering a comprehensive diagnostic approach to early-stage AD.

This study centers on two primary pathological hallmarks of AD:Amyloid beta (Aβ) plaques;Neurofibrillary tangles (NFTs).

From a computational standpoint, this research explores the *Time Factor Hypothesis*, which postulates that early detection, quantification, and intervention targeting neuronal biomarker alterations—potentially decades before clinical manifestation—could facilitate timely diagnosis and effective preventative strategies. The hypothesis advocates screening younger adults to identify early biomarker deviations, representing a potentially transformative approach to combating AD onset and progression. In the absence of a cure, early detection enables prompt diagnosis and the timely initiation of preventive measures, helping to avoid the expense of prolonged patient suffering, patient care, and management.

Ultimately, elucidating the age-related progression dynamics of AD biomarkers may significantly enhance diagnostic precision and inform the development of targeted therapeutic interventions. Our findings underscore the critical importance of initiating early interventions, potentially as early as ages 30 to 40, to substantially reduce both the prevalence and severity of Alzheimer’s disease.

## 2. Computational Framework

This section presents the computational framework developed to model cognitive impairment progression and assess Alzheimer’s disease (AD) risk in younger adults. Grounded in the *Time Factor Hypothesis*, the model is designed to capture the non-linear trajectory of biomarker changes leading to cognitive decline, potentially decades before clinical symptoms manifest. The conceptual diagram for this fit is represented in Figure 1 starting from Biomarker Data → Simulation → Risk Zones → Correlative Risk Scoring → Risk Classification.

**Axiom**: The framework is based on gradual pathological changes in the brain.

The gradual pathological accumulations follow a non-linear progression—starting subtly, accelerating over time, and eventually plateauing. This biological behavior is best represented by a sigmoid function, which allows the model to simulate early-stage deviations in biomarkers before cognitive impairment becomes clinically apparent.

The model incorporates the following key biomarkers, each weighted based on its relative contribution to cognitive risk:Cerebrospinal fluid (CSF) Aβ42;Amyloid PET imaging;CSF Tau and phosphorylated Tau (p-Tau);MRI FDG-PET (brain metabolism).

The computational framework is structured into three main components:

**Descriptive Analysis**—Under this cadre, we performed a descriptive analysis to establish the expected physiological ranges for the cognitive risk associated with each biomarker. These baseline values were modeled using a sigmoid function to generate a synthetic dataset that captures the biomarker’s variability over time and across age groups.

**Parameter Accumulation**—This component tracks the progression and accumulation of biomarkers over time. By modeling these trajectories, we assessed deviations from normal levels, providing insights into the temporal dynamics of each biomarker concerning AD risk.

**Correlation and Classification**—We analyzed the correlation between biomarker accumulation and neuronal changes associated with Alzheimer’s disease. This enabled us to classify cognitive risk into distinct categories—*normal*, *mild risk*, or *high risk*—based on biomarker fluctuations and their combined effect on *cognitive impairment* (CI) scoring.

This structured computational approach provides a quantifiable framework for identifying early biomarkers of AD and refining predictive models for early diagnosis and risk stratification.

### 2.1. Descriptive Analysis

Previous studies [29,30,31,32,33] have evaluated the expected average levels of key CSF biomarkers across different age groups and populations, including individuals living with HIV infection. Based on these findings, CSF Aβ42 levels below 480 pg/mL or above 800 pg/mL are considered clinically significant indicators of cognitive health status. Specifically, reduced Aβ42 levels suggest amyloid plaque accumulation, while elevated levels are typically associated with normal cognitive function.

For CSF Tau, age-specific thresholds have been proposed: levels should remain below 300 pg/mL for individuals aged 21 to 50, below 450 pg/mL for those aged 51 to 70, and under a critical threshold in individuals aged 70 to 90. Similarly, in amyloid PET imaging, a Centiloid score of 0 is typical in younger adults, while scores approaching 100 are indicative of mild neurodegenerative changes [29,30,31,32,33].

Further research shows that individuals with CSF Aβ42 levels between 600 and 800 pg/mL generally maintain normal cognitive function, whereas levels below 480 pg/mL are linked to progressive cognitive decline [8]. Additionally, [21,34] observed that amyloid PET values less than 7 and CSF Tau levels below 7 are commonly found in cognitively normal individuals. In contrast, amyloid PET values exceeding 7–10 correlate with amyloid positivity and an increased risk of mild cognitive impairment (MCI) and Alzheimer’s disease. Tau levels above the 7–10 range are also associated with early neurodegenerative processes.

CI score further contextualizes these biomarkers, with scores below 3 (CI≤3) associated with normal cognitive aging and scores above 6 indicating early-stage cognitive impairment [34]. Integrating these biomarker thresholds with cognitive impairment scoring provides a structured and quantifiable framework for classifying individuals into cognitive risk zones relevant to Alzheimer’s disease onset and related neurodegenerative disorders.

### 2.2. Sigmoid Simulation and Parameter Accumulation

To standardize the accumulation of biomarker values and imaging results relative to age, we modeled the biomarker measurements using a sigmoid function, defined in Equation (Equation 1). In this formulation, *L* represents the maximum potential value of a given parameter, while *k* serves as a scaling factor to adjust for variability in the input data. The term (x) denotes the individual biomarker measurement, and x0 represents the mean of the respective biomarker column, acting as a reference point for standardization. The sigmoid function is mathematically expressed as:(1)S(x)=L1+ek(x−x0)

This function effectively constrains the output between 0 and *L*, making it well suited for classification tasks where the goal is to assess the likelihood of an individual belonging to a specific cognitive risk category. Within the context of this research, the sigmoid function enables the stratification of individuals into *normal*, *mild-risk*, and *high-risk* groups based on their biomarker profiles associated with Alzheimer’s disease.

To further analyze biomarker progression, we computed the derivative of the sigmoid function, S′(x), to identify critical points and ensure smooth curve behavior. By leveraging regression analysis alongside the derivative S′(x), we reverse-engineered feature distributions, allowing for the controlled generation of synthetic instances representing individuals aged 10 years and older. This approach enriched the dataset, supporting the modeling of early biomarker changes potentially preceding clinical symptoms (see Figure 1 for reference).

Henceforth, the term **donor** may be written as *donor* to reflect the enrichment and synthetic extension of the original dataset.

### 2.3. The Dataset

Most existing datasets in Alzheimer’s research predominantly comprise data from older individuals, typically aged 50 years and above. However, due to the scarcity of available data for younger individuals, particularly those aged 30–50, and the defined nature of available biomarker data —often either *MRI imaging* or *numerical with categorical values*, but rarely both— this study adopts a hybrid dataset approach to enrich the dataset and broaden age representation.

The dataset was constructed by aggregating 307 instances from sources including Kaggle, ANDI, and OASIS. Rigorous data cleaning and filtering procedures were applied to retain relevant features. Duplicate or similar entries were identified and subsequently grouped, with averaged values calculated, using **Age** as the primary instance identifier. To manage missing data, especially for individuals younger than 50, synthetic data points were generated employing the sigmoid-based simulation detailed in Equation (Equation 1).

Table 1 presents a snapshot of the resulting biomarker dataset designed for Alzheimer’s disease (AD) risk assessment. Notably, the dataset starts from age 10, reflecting an intentional focus on early-stage biomarker progression rather than traditional cohorts limited to older populations.

Initial observations suggest that CSF Aβ42 levels increase with age during early development, potentially reflecting normal physiological changes before the expected decline associated with AD. Similarly, amyloid PET and CSF Tau levels demonstrate gradual increases, indicating progressive biomarker changes that may begin well before clinical symptoms emerge.

The dataset includes the following key attributes:**Age**: The individual’s age (beginning at 10 years).**CSF Aβ42**: Cerebrospinal fluid amyloid beta 42 levels, a biomarker indicating amyloid plaque accumulation, a hallmark of AD.**Amyloid PET**: Positron emission tomography measurements of amyloid deposition in the brain, where higher values denote greater amyloid accumulation.**CSF Tau**: Levels of Tau protein in cerebrospinal fluid, serving as an indicator of neurodegeneration associated with AD.**MRI FDG-PET**: A neuroimaging metric capturing structural and metabolic brain changes.

This enriched dataset enables the investigation of biomarker dynamics across a broader age range, offering valuable insights into early-stage Alzheimer’s risk assessment.

### 2.4. Correlation Analysis and Cognitive Risk Categories

It is critical to identify correlations between biomarkers and establish classification regions that stratify *donors* into *normal (no_risk)*, *mild_risk*, and *high_risk* cognitive categories associated with AD. For instance, the significance of CSF Aβ42 is that its reduction signals amyloid plaque accumulation in the brain, a hallmark of Alzheimer’s disease (AD) [30].

Furthermore, Figure 2 and Figure 3 illustrate two key relationships in the dataset. Figure 2 presents the age-dependent trajectory of CSF Aβ42 levels, displaying a sigmoidal trend. CSF Aβ42 levels rise gradually during early life (ages 10–30), possibly reflecting normal amyloid metabolism. From midlife (ages 30–60), Aβ42 levels increase more rapidly, potentially indicating changes in amyloid clearance efficiency. Levels plateau in later years (60+), likely due to reduced clearance or plaque accumulation in brain tissues.

This trajectory aligns with established AD biomarker models, where CSF Aβ42 concentrations decline in individuals with amyloid pathology [16,29,35].

Figure 3 on the other hand, highlights the correlation between CSF Aβ42 and CSF Tau levels. The scatter plot and regression line reveal an inverse relationship: CSF Tau increases sharply as CSF Aβ42 decreases. This supports the hypothesis that amyloid deposition (low Aβ42) is linked to neurodegeneration (elevated Tau), both of which are critical to AD progression.

Applying predefined medical thresholds, Section 2.1 provides clear cutoffs for biomarker levels, enabling the classification of cognitive states (Figure 4). This visualization traces biomarker trajectories across the lifespan, highlighting cognitive risk zones. The shaded backgrounds (green, yellow, orange, and red) indicate transitions from normal cognitive function to mild cognitive impairment. With increasing age, deviations in biomarker levels become more pronounced, particularly in individuals transitioning into high-risk or MCI categories.

These cognitive risk zones are defined as:**Normal:** Biomarker levels within safe physiological ranges;**Mild Risk:** Slightly elevated biomarker levels indicating potential early changes;**High Risk:** Significant biomarker abnormalities but without formal clinical diagnosis;**MCI (Mild Cognitive Impairment):** Biomarker levels exceed critical thresholds indicating cognitive deterioration.

As illustrated, CSF Aβ42 (purple) declines sharply with age, while amyloid PET (red), CSF Tau (yellow), and MRI + FDG PET (blue) show progressive increases. The green cognitive impairment curve mirrors this upward trend, reinforcing the relationship between biomarker deviations and cognitive decline.

Figure 5 models the progression of cognitive impairment as a function of age, segmented into cognitive risk categories. Data points, color-coded by risk level, reveal a nonlinear increase in cognitive impairment over time. Initially, most individuals remain in the normal range (blue). As age advances, the probability of transitioning into mild risk (orange), high risk (green), and MCI (red) increases significantly.

The trajectory indicates a critical period around midlife (50–60 years), where cognitive risk accelerates sharply. This observation aligns with neurodegenerative models suggesting that biological and cognitive reserves initially buffer against decline until cumulative damage leads to rapid deterioration.

These findings emphasize the importance of early detection and monitoring. Individuals classified as mild-risk still represent a key intervention window where preventive strategies could delay or mitigate progression. Integrating machine learning techniques could further enhance this model by identifying subtle early indicators of cognitive impairment.

Considering Figure 4 and Figure 5, the central research question emerges: *What biomarker values can individuals aged* 30–40 *(and* 40–50*) maintain to remain within the safe (green) zone for healthy AD-free old age? Or put it in another way, what thresholds signal progression toward mild or high risk of developing Alzheimer’s disease at an older age?*

## 3. Computational Summation

To answer the final question in the previous section, it is essential to recall and reaffirm the objective of this paper. To achieve this, we restrict and define the potential symptoms of brain atrophy based on cognitive impairment observed in *donors*.

### Cognitive Impairment

Cognitive impairment (CI) was modeled as a weighted sum of biomarker values (CSF Aβ42, amyloid PET, CSF Tau, MRI FDG PET), incorporating an exponential scaling factor to account for the accelerated impairment. Since biomarkers contribute (approximately) linearly to cognitive impairment, an increase in amyloid PET and CSF Tau elevates the risk, while a decrease in CSF Aβ42 also contributes to increased risk. The terms can be modeled asB1(A)=k1e−m1A,B2(A)=k2e−m2A,B3(A)=k3e−m3A,B4(A)=k4e−m4A
where B1(A) is the CSF Aβ42 value, which decreases exponentially with age. B2 corresponds to the amyloid PET value, which increases exponentially with age. B3(A) represents (CSF Tau), and B4(A) denotes MRI FDG PET, both of which increase at different rates over time, such that *cognitive impairment*(CI) can be rewritten as:(2)CI(A)=α.eβA+w1k1e−m1A+w2k2e−m2A+w3k3e−m3A+w4k4e−m4A

Thus, we define the risk zones—*safe, mild risk, and unsafe*—using Equation (Equation 2), based on the values of the underlying biomarker parameters. Specifically, this equation determines the range of biomarker values an instance aged 30–40 (30–50) must possess to remain safe, be at mild risk, or become unsafe from the disease at an older age. Specifically, the following system of exponential equations explains it succinctly: (3)SafeZone:CI(A)=α.eβA+∑iwi.Bi(A)≤γ(4)MildRiskZone:γ<α.eβA+∑iwi.Bi(A)≤δ(5)UnsafeZone:α.eβA+∑iwi.Bi(A)>δ
where *A* represents age, CI(A) denotes the cognitive impairment score at age *A*, and Bi(A) corresponds to the biomarker level at age *A*. The parameter wi is the weight coefficient that determines the contribution of each biomarker value to cognitive impairment. The parameters α and β are scaling factors of CI(A) [36], where cognitive decline is modeled by the exponential function eβA, capturing the accelerated progression of impairment with aging.

Safe zone holds when CSF Aβ42 dominates over other biomarkers and remains high, and amyloid PET, CSF Tau, and MRI FDG PET remain below mild risk levels. A mild zone occurs if CSF Aβ42 declines slightly, but amyloid PET and CSF Tau rise moderately (perhaps accounting for the early onset of AD). The evidence for an unsafe zone is given by a significant drop of CSF Aβ42 and amyloid PET, CSF Tau, and MRI FDG PET reach high threshold levels. Referencing Section 2.1, we conclude that γ=3 and δ=6 within the 30–40 age bracket and 4 and 7 for 40–50 years age bracket, respectively.

## 4. Model Output and Cognitive Risk Zone Classification

We leveraged ChatGPT 4o [37] to generate a synthetic dataset containing 150 instances for testing (due to the challenge of acquiring heterogeneous real-world datasets) and computed the cognitive impairment score for each instance using Equation (Equation 2). Each instance was then classified into its respective cognitive risk zone based on the thresholds defined in Equations (Equation 3)–(Equation 5), with an example of the classification summarized in Table 2.

The model output produced age-specific CI risk bands (Table 3) guided by strict threshold templates (Table 4, Table 5 and Table 6). Notably, risk classification dynamically references the age-specific intervals during CI analysis—failure to do so would result in overestimating risk in younger individuals and underestimating it in older ones.

The 30–40 age group applies the strictest thresholds, reflecting the expectation of relatively healthy biomarkers. In contrast, the 40–50 age group tolerates mild biomarker deviations, shifting the “Safe” range upward. The 50–60 age group follows a similar trend, reflecting normal biomarker drift with aging.


**Key observations include the following:**
Higher CSF Aβ42 levels are protective, reducing the cognitive risk score.Elevated amyloid PET and Tau levels increase the CI score, indicating greater neurodegenerative risk.Age contributes exponentially, but moderately, due to the model’s exponential scaling component.


Notably, MRI scan results can reveal structural brain changes, including mild parietal or temporal lobe atrophy, cortical thinning, white matter lesions, or hippocampal volume loss. The model identified 17 instances classified within the mild-risk zone. Next, we examine six representative mild-risk cases to gain deeper simultaneous insights into biomarker patterns and cognitive risk profiles in the next section.

Our model further supports retrospective analysis of disease trajectories, providing estimates of the optimal intervention window. This is particularly valuable for aged *donors* already exhibiting Alzheimer’s disease (AD) symptoms, helping to infer when earlier intervention might have slowed disease progression. In such cases, this perhaps serves as evidence in support of further aggressive medical attention.

### Discussion of Table 2: Sample Classification Results of Selected Instances with CI Scores and Risk Zones

Table 2 presents the classification of eight sampled instances based on their biomarker values, computed cognitive impairment (CI) scores, and assigned cognitive risk zones according to the model’s age-specific thresholds. Key Observations are:Higher CSF Aβ42 values are associated with lower cognitive impairment (CI) scores and safer classifications. For instance, the 53-year-old individual exhibits the highest CSF Aβ42 level (516.39 pg/mL), which corresponds to a low CI score of 3.21 and a Safe classification, despite moderate Tau levels. This observation highlights the protective role of CSF Aβ42, where higher levels reduce cognitive risk, while lower CSF Aβ42 levels are indicative of amyloid plaque formation and increased risk of Alzheimer’s disease.The 40-year-old instance exhibits high amyloid PET (19.13) and elevated MRI FDG-PET (14.58) values. Despite being relatively young, this drives its CI to 3.37, placing it in the mild risk zone due to age-specific stricter thresholds for younger adults. Similarly, higher CSF Tau levels (e.g., 411.34 at age 47) contribute to mild risk classification, reiterating that elevated amyloid PET and Tau drive risk upward.Notice the *age-specific sensitivity* to biomarker deviations. For example, the 34- and 39-year-old instances fall into the mild risk zone even with moderate biomarker levels. This reflects the model’s stricter criteria for younger adults, where any abnormal biomarker deviation is penalized more heavily. Older individuals (47–58) tolerate similar or higher biomarker values but remain classified as safe or mild risk due to relaxed thresholds.Elevated MRI FDG-PET (metabolic changes) values are seen in mild risk cases, especially in the 40-year-old (14.58), suggesting early structural/metabolic brain changes contributing to increased cognitive risk.Overall, safe cases maintain CI below 4 (Ages 58 and 53). Mild risk cases occupy the CI range of 3.37 to 5.35, with no instances classified as unsafe in this sample.

## 5. Discussion: Limitations and Result Interpretation

### 5.1. Limitations

To address sparse biomarker data in adults under 50, we generated synthetic instances using a sigmoid-based simulation. While this method captures plausible non-linear trajectories, it may not fully reflect the true variability or covariation patterns present in real-world cohorts. As a result, our age-specific risk thresholds—particularly in younger age brackets—could be overly simplified, optimistic, or fail to account for rare but critical deviations. Future validation on longitudinal clinical datasets is essential to confirm that the synthetic profiles do not bias the model toward underestimating early pathology.Our framework relies on cross-sectional snapshots of biomarker levels across ages rather than following the same individuals over time. This limits our ability to capture intra-individual change dynamics and may conflate cohort effects (e.g., generational differences in risk factors) with true aging trajectories. Consequently, the risk-zone boundaries derived here may shift when applied to actual longitudinal data, potentially altering the predicted optimal intervention windows.Each core biomarker—CSF Aβ42, amyloid PET, CSF Tau, and FDG-PET—has inherent technical variability: assay differences, scanner calibration, and inter-laboratory protocols can introduce measurement noise or systematic offsets. Although we modeled thresholds broadly, unaccounted measurement noise could widen confidence intervals around our CI scores, reducing the precision of individual risk stratification and potentially increasing false positives or negatives. We were also not involved in the data acquisition processes.Our current model weights only biochemical and imaging biomarkers, omitting key genetic (e.g., APOE ϵ4 status) and lifestyle (e.g., exercise, diet) modifiers of AD risk. This simplification may overlook synergistic effects, such as how physical activity might mitigate Tau-related risk, thereby limiting predictive power and potentially misclassifying individuals whose biomarker profiles interact strongly with non-biological factors.We selected a standard sigmoid to model biomarker accumulation and decline. However, real-world biomarker trajectories may deviate, exhibiting multi-phase kinetics, plateaus, or even transient regressions. Relying on a fixed functional form constrains the model’s flexibility and could misestimate early or late-life risk slopes, affecting downstream CI calculations and the timing of recommended screenings.All results stem from externally generated or synthetic datasets. Without testing on independent cohorts—especially those with diverse genetic backgrounds, comorbidities, and imaging platforms—our thresholds remain provisional. External validation is crucial both for confirming generalizability and for recalibrating CI cut-points to local populations or different imaging centers.We categorize risk into four discrete zones (normal, mild, high, MCI) based solely on CI cut-points. In practice, AD progression is a continuum. Discrete zoning may overlook subthreshold changes and fail to capture personal trajectories that straddle boundary regions. More granular or individualized probabilistic risk scores could provide finer guidance for clinicians.

### 5.2. Limitations Impact on Results

Collectively, these limitations imply that our absolute risk estimates and age-specific thresholds should be interpreted cautiously. Synthetic augmentation and cross-sectional design may bias early-age risk downward, potentially delaying necessary interventions. Measurement variability and missing non-biological factors could reduce predictive accuracy, leading to misclassification in a minority of cases. Finally, without external validation, the model’s translational applicability remains unproven. Addressing these limitations in future work, through longitudinal cohort studies, richer feature integration, flexible modeling approaches, broader discussions, and rigorous external benchmarking, will be essential to refine our computational tool and ensure reliable, personalized risk stratification in clinical practice.

## 6. Conclusions

This research presents a computational modeling framework designed to explore and predict the early onset of Alzheimer’s disease (AD) in younger adults, decades before the manifestation of clinical symptoms. By leveraging sigmoid-based data generation and cognitive impairment modeling, the study simulates non-linear biomarker trajectories that align with known patterns of disease progression.

Through the development of age-adjusted cognitive risk zones, the model provides a quantifiable system for classifying individuals based on biomarker deviations and their impact on cognitive decline. Notably, higher CSF Aβ42 levels consistently demonstrated a protective effect, while elevated amyloid PET and Tau levels correlated with increased cognitive impairment scores. These findings underscore the importance of integrating age-specific thresholds to prevent over- or under-estimation of cognitive risk across different age groups.

A key contribution of this work is its focus on pre-symptomatic risk detection using computational methods. The framework enables researchers to model early disease trajectories and offers clinicians a tool for identifying individuals who may benefit from timely preventive interventions. Importantly, the model’s flexibility allows for future expansion to incorporate genetic factors, neuroimaging metrics, and lifestyle variables, enhancing its applicability in clinical settings.

Future directions include validating the framework with longitudinal real-world datasets, enhancing predictive accuracy using machine learning architectures such as recurrent neural networks, and applying explainability tools to interpret feature contributions. Additionally, integrating digital twin simulations could further personalize risk assessments and support precision treatment planning.

Overall, this research demonstrates the potential of computational modeling to bridge the gap between complex biomarker dynamics and early diagnostic opportunities in Alzheimer’s disease. By establishing a robust mathematical and statistical foundation, this work contributes to advancing precision medicine approaches in neurodegenerative disease research and clinical care.

## Figures and Tables

**Figure 1 diagnostics-15-01327-f001:**
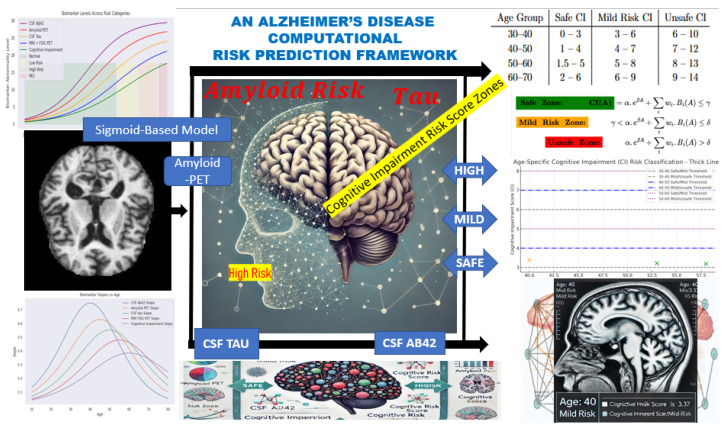
Conceptual diagram illustrating the computational framework for Alzheimer’s disease risk prediction. It captures Biomarker Data → Simulation → Risk Zones → Correlative Risk Scoring → Risk Classification.

**Figure 2 diagnostics-15-01327-f002:**
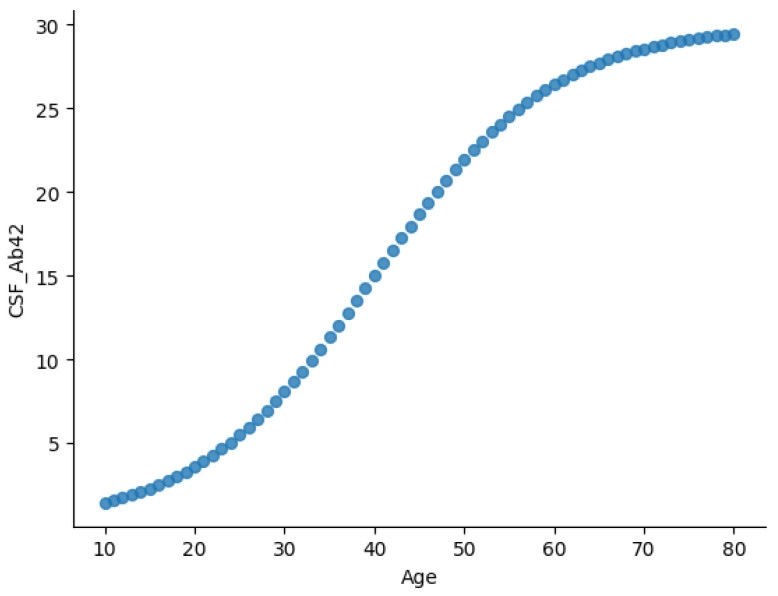
Age-dependent sigmoidal increase in CSF Aβ42, reflecting key transitions in amyloid metabolism over time.

**Figure 3 diagnostics-15-01327-f003:**
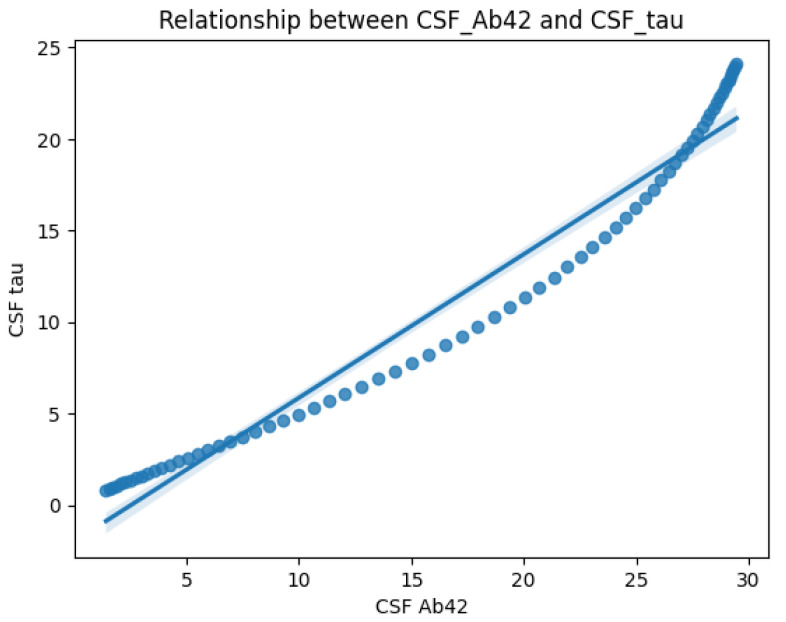
Age-dependent correlation plot showing that elevated Tau levels are associated with reduced Aβ42, supporting amyloid-driven neurodegeneration in AD.

**Figure 4 diagnostics-15-01327-f004:**
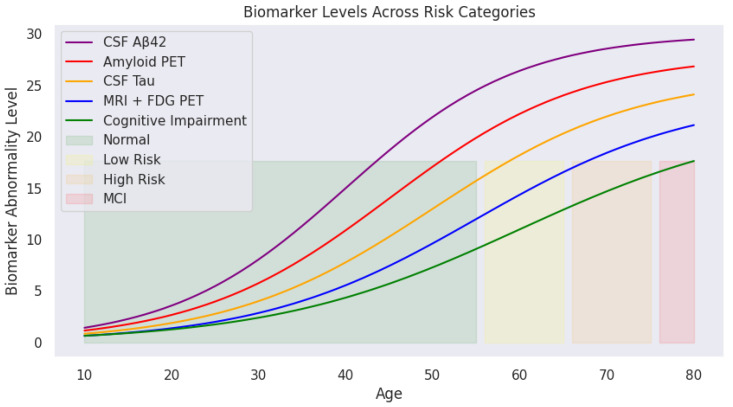
Age-related biomarker trajectories transformed by sigmoid scaling. CSF Aβ42 (purple) declines with age, while amyloid PET, CSF Tau, and MRI + FDG PET levels rise, reflecting the progressive risk of cognitive impairment. Shaded areas indicate transitions between cognitive risk categories.

**Figure 5 diagnostics-15-01327-f005:**
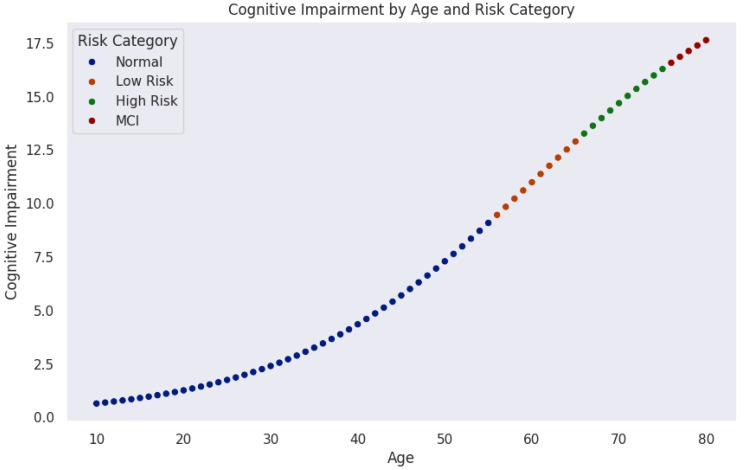
Cognitive impairment progression over age, color-coded by risk levels. The model shows an initial period of cognitive stability, followed by gradual and then accelerated increases in risk, particularly after age 50.

**Table 1 diagnostics-15-01327-t001:** Early biomarker progression and cognitive assessment in young individuals. This table presents the biomarker levels for individuals. Ages 10 to 110 years were considered, focusing on early-stage Alzheimer’s disease (AD) risk factors. The dataset includes key cerebrospinal fluid (CSF), neuroimaging biomarkers such as CSF 
$A\beta_{42}$
, amyloid PET, CSF Tau, and MRI FOG PET.

Age	CSF_Ab42	Amyloid PET	CSF Tau	MRI FOG PET
10	1.422776195	1.15055579	0.839682182	0.63837886
11	1.546666892	1.251055685	0.91156436	0.689563902
12	1.719772277	1.36639224	0.989358424	0.74564352
13	1.88920682	1.488831819	1.073507365	0.805661359
14	2.0745661	1.60277478	1.164480779	0.878329193
15	2.225745401	1.76235397	1.262772013	0.033977347
16	2.45188095	1.197938204	1.368893079	1.014954524
17	2.73368883	2.085104465	1.483419148	1.055688099
18	2.995514674	2.265577082	1.606890123	1.188384416
19	3.272046536	2.400189615	1.739908743	1.275627274
20	3.576887651	2.666988017	1.883888619	1.375780022
21	3.003254331	2.885211264	2.037061649	1.48328319
22	4.255311947	3.13731708	2.202474709	1.598592854
23	4.633957953	3.396927461	2.379885564	1.722181014
24	5.03344846	3.6748514	2.570257903	1.885453881
25	5.72765714	3.771898817	2.773955443	1.996147716
26	5.934483343	4.288584041	2.991735033	2.147521424
27	6.4249505099	4.625735365	3.224238726	2.309173001

**Table 2 diagnostics-15-01327-t002:** Sample classification results of selected instances with CI scores and risk zones.

Age	CSF $A\beta_{42}$	Amyloid_PET	CSF_Tau	MRI_FDG_PET	CI	Risk Level
58	428.21	14.28	382.99	6.92	3.18	Safe
40	411.53	19.13	370.31	14.58	3.37	Mild Risk
53	516.39	8.62	478.00	3.19	3.21	Safe
39	413.08	14.25	403.98	6.57	4.41	Mild Risk
34	422.18	15.37	393.98	6.55	3.62	Mild Risk
47	427.23	14.69	411.34	6.29	5.30	Mild Risk
48	419.70	14.01	414.07	6.95	4.21	Mild Risk
50	418.06	15.26	399.28	7.18	5.35	Mild Risk

**Table 3 diagnostics-15-01327-t003:** Age-specific cognitive impairment (CI) risk bands used for model classification.

Age Group	Safe CI	Mild Risk CI	Unsafe CI
30–40	0–3	3–6	6–10
40–50	1–4	4–7	7–12
50–60	1.5–5	5–8	8–13
60–70	2–6	6–9	9–14

**Table 4 diagnostics-15-01327-t004:** Estimated biomarker thresholds for ages 30–40.

Biomarker	Normal (Green)	Low Risk (Yellow)	High Risk (Orange)
CSF_ $A\beta_{42}$	(15, 22)	(10, 15)	(5, 10)
Amyloid_PET	(4, 7)	(7, 10)	(10, 15)
CSF_Tau	(3, 6)	(6, 9)	(9, 12)
MRI_FDG_PET	(2, 5)	(5, 8)	(8, 12)
Cognitive_Impairment	(0, 3)	(3, 6)	(6, 10)

**Table 5 diagnostics-15-01327-t005:** Estimated biomarker thresholds for ages 40–50.

Biomarker	Normal (Green)	Low Risk (Yellow)	High Risk (Orange)
CSF_ $A\beta_{42}$	(12, 18)	(8, 12)	(5, 8)
Amyloid_PET	(5, 9)	(9, 12)	(12, 18)
CSF_Tau	(4, 7)	(7, 10)	(10, 14)
MRI_FDG_PET	(3, 6)	(6, 9)	(9, 13)
Cognitive_Impairment	(1, 4)	(4, 7)	(7, 12)

**Table 6 diagnostics-15-01327-t006:** Estimated biomarker thresholds for ages 55–65.

Biomarker	Normal (Green)	Low Risk (Yellow)	High Risk (Orange)
CSF_ $A\beta_{42}$	(10, 15)	(6, 10)	(3, 6)
Amyloid_PET	(7, 12)	(12, 16)	(16, 22)
CSF_Tau	(5, 8)	(8, 12)	(12, 16)
MRI_FDG_PET	(4, 7)	(7, 10)	(10, 15)
Cognitive_Impairment	(2, 5)	(5, 9)	(9, 14)

## Data Availability

Link to data sources of the datasets used: https://adni.loni.usc.edu/data-samples/adni-data/ (accessed on 23 February 2025); https://www.kaggle.com/discussions/accomplishments/510182 (accessed on 23 February 2025); https://www.kaggle.com/datasets/jboysen/mri-and-alzheimers (accessed on 23 February 2025).

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
