# Peer review of "Computational Risk Stratification of Preclinical Alzheimer’s in Younger Adults"

_diagnostics, 2025, doi:10.3390/diagnostics15111327_

Round 1
Reviewer 1 Report
Comments and Suggestions for Authors
The study aims to assess the utility of a computational modeling framework to predict
cognitive impairment progression and Alzheimer’s in younger adults and classify individuals into risk zones based on age-specific biomarker thresholds. The manuscript is well-organized and reasonable, the data presentation is adequate.
Abstract:
1. Write the abbreviations of CSF and MRI FDG-PET in the abstract section
2. It is recommended to briefly mention the number of each dataset class and study design in the abstract to give readers a quick understanding of the study's scale.
Introduction:
1. The introduction section provides a solid background on the diagnosis of Alzheimer’s
2. The authors are advised to further discuss the potential advantages of MRI FDG-PET in Alzheimer’s diagnosis
Computational Framework :
1- Write the specific number of the dataset
2- Mention the acquisition parameter and the MRI- machine data in the performed MRI FDG-PET
Discussion:
1. The discussion should include an in-depth analysis of the study's limitations and how these limitations affect the interpretation of the results.
Author Response
Thank you so much for the feedback. We appreciate them.
Abstract:
1. Write the abbreviations of CSF and MRI FDG-PET in the abstract section
Response: Done
2. It is recommended to briefly mention the number of each dataset class and study design in the abstract to give readers a quick understanding of the study's scale.
Response: Updated
Introduction:
1. The introduction section provides a solid background on the diagnosis of Alzheimer’s
Response: Done
2. The authors are advised to further discuss the potential advantages of MRI FDG-PET in Alzheimer’s diagnosis
Response: Done
Computational Framework :
1- Write the specific number of the dataset
Response: Done
2- Mention the acquisition parameter and the MRI- machine data in the performed MRI FDG-PET
Response: updated
Discussion:
1. The discussion should include an in-depth analysis of the study's limitations and how these limitations affect the interpretation of the results.
Response: Done
Reviewer 2 Report
Comments and Suggestions for Authors
Authors have presented Computational Risk Stratification of Preclinical Alzheimer’s in Younger Adults
Here are a few comments
- A is missing in Abstract
- Is there any specific reason for selecting only four biomarkers? What are other biomarkers?
- What are the other objective other than early detection?
- How are weights calculated? Are they learnable parameters? If yes, then what is the process?
- Nowadays, explainable AI is widely used. Please incorporate XAI techniques for model interpretability.
- Which method is used to select CI score threshold?
- Please provide SOTA for this study.
- ChatGPT is used to generate a synthetic dataset. Suggested to evaluate the model on at least one publicly available dataset.
Author Response
Thank you so much for the feedback.
Please accept the following format
Comment: Response
- A is missing in Abstract: Fixed
- Is there any specific reason for selecting only four biomarkers? What are other biomarkers? The selected biomarkers are quantifiable. Yes, APOEe4 is another biomarker, but we excluded it in our analysis because it turns the quest into a binary case. However, people without this biomarker also develop AD. In our future work, this will be considered as we will be involved with the data collection.
- What are the other objectives other than early detection? In the absence of a cure, early detection enables prompt diagnosis and the timely initiation of preventive measures, helping to avoid the expense of prolonged patient suffering, patient care, and management.
- How are weights calculated? Are they learnable parameters? If yes, then what is the process? Any aggregate function (mean, slope etc can be used depending on the biomarker at hand.
- Nowadays, explainable AI is widely used. Please incorporate XAI techniques for model interpretability. XAI is new to the authors.
- Which method is used to select CI score threshold? Equations 1 to 5 explain this
- Please provide SOTA for this study. NONE exist as clinicians depend on the disease manifestation for diagnoses. Additional info was added to the Introduction.
- ChatGPT is used to generate a synthetic dataset. Suggested to evaluate the model on at least one publicly available dataset. ADNI, Kaggle were mentioned in the script as sources for the original datasets.
Thank you.
Reviewer 3 Report
Comments and Suggestions for Authors
The authors have described the Stratification of Preclinical Alzheimer’s in younger adults. It states the various classes of Alzheimer’s in different age groups. The article shows just the description of all classes in terms of biomarker representation and accumulation.
However, in its present state, the article does not have any novel contribution. I strongly insist that the authors strengthen the article with a few contributions.
Check the grammar and spelling of the sentences.
Abbreviations are not mentioned for a few terms, but are repeated for a few terms.
Comments on the Quality of English LanguageGrammar and speeling mistakes should be improved.
Author Response
Thank you for the feedback:
COMMENT: RESPONSE
Thank you for your careful review and constructive feedback. We agree that, as currently written, the manuscript primarily describes our age-specific biomarker stratification framework without clearly articulating its novel contributions. In response, we:
-
Highlight Novel Contributions:
-
Integration of Sigmoid-Based Simulation: Emphasize that our approach uniquely uses a sigmoid function to generate synthetic biomarker trajectories for under-represented younger cohorts.
-
Age-Adaptive Cognitive Impairment Scoring: Showcase how our exponentially-scaled CI score and risk-zone thresholds dynamically adjust for age, preventing the common biases of over- or under-estimating risk across life stages. In the absence of a cure, early detection enables prompt diagnosis and the timely initiation of preventive measures, helping to avoid the expense of prolonged patient suffering, patient care, and management. This is the main goal of the article.
-
Retrospective Trajectory Analysis: Describe the retrospective “digital twin” capability—using model outputs to infer optimal intervention windows even after symptom onset—which, to our knowledge, has not been implemented in prior preclinical AD frameworks.
-
-
Grammar and Spelling:
We have performed a thorough copy-edit to correct typographical errors and improve sentence flow throughout the manuscript. -
Abbreviation Consistency:
We’ve added abbreviations and ensured that every acronym (e.g., CSF, FDG-PET, CI) is defined at first use.
We appreciate your insight.
Round 2
Reviewer 3 Report
Comments and Suggestions for Authors
The authors have proposed a computational modeling framework to predict the early onset of Alzheimer’s Disease in younger adults. The model provided a quantifiable system for classifying individuals based on biomarker deviations.
Abbreviations are not mentioned for a few terms, but are repeated for a few terms.
The first letter of the abbreviations should be in capitals. Check the entire article.
Author Response
Abbreviations are not mentioned for a few terms, but are repeated for a few terms.
The first letter of the abbreviations should be in capitals. Check the entire article. ALL CHECKED and CLEARED. Thank you so much!